# Factors Associated with Frequency of Peanut Consumption in Korea: A National Population-Based Study

**DOI:** 10.3390/nu12051207

**Published:** 2020-04-25

**Authors:** Minyoung Jung, Jayun Kim, Su Mi Ahn

**Affiliations:** 1Department of Pediatrics, Kosin University Gospel Hospital, Kosin University School of Medicine, Busan 49267, Korea; 2Kosin Innovative Smart Healthcare Research Center, Kosin University Gospel Hospital, Busan 49267, Korea; jydk6557@naver.com (J.K.); asm0400@hanmail.net (S.M.A.); 3Department of Nutrition, Kosin Gospel University Hospital, Busan 49267, Korea

**Keywords:** peanut, dietary pattern, education, sociodemographic factor, nutrition, prevention of peanut allergy

## Abstract

Household peanut exposure via skin in infants with impaired skin barrier function is a risk factor for peanut allergy development. The aim of this study is to investigate the peanut consumption of Koreans using national representative data. We used data from the Korean National Health and Nutrition Examination Survey 2012–2016, consisting of data from 17,625 adults who complete the survey. Peanut intake was assessed using a 24-h recall method. Of the study population, 10,552 (59.9%), 6726 (38.2%), and 347 (1.9%) subjects were categorized into non-intake, intermittent intake, and frequent intake group, respectively. Ordered logistic regression models were used to examine the association between sociodemographic and dietary factors and the frequency of peanut intake. After adjusting for confounders, increasing age (adjusted odds ratio (aOR) 1.03; 95% confidence interval (CI) 1.03–1.04), higher education (high school graduates: aOR 1.75, 95 CI 1.39–2.19; higher than college: aOR 2.11, 95% CI 1.65–2.70), and prudent dietary scores in the second (aOR 1.71; 95% CI 1.47–1.99), third (aOR 2.53; 95% CI 2.16–2.97) and the fourth quartiles (aOR 3.72; 95%CI 3.16–4.40) were associated with a high frequency of peanut consumption. This information may be helpful not only in public health research for nutrition but also in personal management for the prevention of peanut allergy in Korea.

## 1. Introduction

Peanut is a legume that belongs to the family of Fabaceae and is botanically named *Arachis hypogaea* [1]. It is usually included among the oilseeds due to its high oil contents [2]. They provide not only a rich source of unsaturated fatty acids, high-quality protein, arginine, vitamin E, and other antioxidants [3], but also value via the phytonutrient composition (carotenoid, stilbenes, lignans, phenolic aldehydes flavonoids, and phytosterols) [1,4,5]. Due to these health benefit properties [6], peanut is recognized to reduce risks of cardiovascular disease (CVD) [7,8,9], complications of diabetes mellitus (DM) [10], cancer [11,12], and cognitive dysfunction [13]. Therefore, peanuts are widely consumed roasted or boiled, and also processed into various forms such as butter, candy, chocolates, cakes, beverages, and others.

While peanuts are healthy food, some people cannot eat them due to allergic responses. Peanuts are one of the major causes of food-allergic reactions, and peanut allergies can lead to fatal anaphylaxis [14,15,16]. Patients with peanut allergy and their families reported a negative impact on their quality of life [17,18,19]. Persistent fear of serious allergic reactions leads to the avoidance of public transport, eating in restaurants, and traveling, as well as reduced school performance [20,21,22]. Because peanut allergy rarely demonstrates intolerance acquisition [23], strategies for the prevention of peanut allergy are important.

Recently, a strategy involved with the introduction to solid foods has been implemented to prevent food allergy. In infants at high-risk for developing an allergy to peanut, the Learning Early About Peanut Allergy (LEAP) trial showed that early peanut introduction at 4 to 11 months reduced the prevalence of peanut allergy at 5 years of age compared to delayed peanut introduction [24]. Consequently, recent guidelines have recommended the early introduction of peanut-containing products in high-risk infants (e.g., those with severe atopic dermatitis and/or an egg allergy). This has resulted in the recent Western guidelines that recommend introducing peanut-containing products early in high-risk infants (e.g., those with severe atopic dermatitis and/or presence of egg allergy) [25,26,27]. However, the prevalence of peanut allergy in Asian countries has ranged from 0.2% to 0.6% [28,29], lower than that in Western countries [30,31]. Moreover, the traditional introduction of complementary foods has not included peanuts for Korean infants. Therefore, the applicability of the Western guidelines to Korean infants is unclear [32].

However, the dual-allergen exposure hypothesis postulates that environmental exposure to peanuts through a disrupted skin barrier rather than through oral exposure may be an important pathogenic route in infants at high risk of food allergy [33,34]. The Swedish population birth cohort study found that high household peanut exposure in early childhood may increase the risk of peanut sensitization in atopic children [35]. Therefore, household peanut exposure may be a risk factor for peanut allergy in infants with impaired skin barrier function [36]. Peanut consumption varies worldwide [3]. However, there are no data about peanut consumption in the Korean population. To provide baseline information on household peanut consumption, we aim to investigate the characteristics of household peanut consumption in Koreans using national representative data. 

## 2. Materials and Methods 

### 2.1. Study Design and Populations 

We used data from the Korean National Health and Nutrition Examination Survey (KNHANES), a nationwide cross-sectional survey that has been used to assess the health and nutritional status of the Korean population. A survey for health status and dietary intake, as well as the performance of health examinations, were conducted annually by the Korean Center for Disease Control and Prevention (KCDC). Stratified multistage clustered probability sampling was used to obtain nationally representative data. In the present study, 17,625 subjects aged over 19 years who completed the KNHANES dietary intake survey between 2012 and 2016 were included (Figure 1). This study was approved by the Institutional Review Board of Kosin University Gospel Hospital (IRB file No. KUGH 2019-12-036). Informed consent was not required as the claim databases used in the present study contain only de-identified data.

### 2.2. Data Variables

Data on sociodemographic variables data, including age, gender, levels of education, household incomes, occupation, household composition, smoking status, and alcohol consumption, were collected during face-to-face interviews. Level of education was divided into four groups of elementary, middle school, high school, and college or higher. Level of household income was categorized as low, lower–middle, higher–middle, and high according to quartile of monthly family income. Occupation was originally categorized into 10 groups (managers, professional, clerical workers, service workers, sales workers, agricultural and fishery workers, skilled workers, craft workers, elementary workers, military, and unemployed) based on the Korean Standard Classification of Occupations of the Korea National Statistical Office. In the present study, occupations were reclassified into four categories (professional and manager, non-manual, skilled, unskilled workers, and unemployed). Current smoking status was classified as never (never smoked cigarettes in lifetime), former (smoked cigarette in lifetime but current non-smoker), and current (current smoker). Alcohol consumption was defined as the frequency of alcoholic beverages consumed in the past 12 months and categorized as none (no consumption of any type of alcoholic beverage), moderate (≤4 times per week), and heavy (>4 times per week). The body mass index (BMI) was measured as weight in kilograms divided by square of height in meter. BMI was classified into underweight (BMI under 18.5 kg/m^2^), normal (BMI greater than or equal to 18.5 to 24.9 kg/m^2^), overweight (BMI greater than or equal to 25 to 29.9 kg/m^2^), and obesity (BMI greater than or equal to 30 kg/m^2^) according to World Health Organization guidelines [37]. CVD was defined as at least one of hypertension, ischemic heart disease, hypercholesterolemia, or stroke. Data on the history of DM was also collected.

Dietary intake was assessed using a 24-h recall method with a trained interviewer-administered semiquantitative food frequency questionnaire (FFQ) containing 122 items. For peanut consumption, participants were asked how often they consumed peanuts during the last year. The selected answers were: “(1) I rarely consume peanuts”, “(2) once per month”, “(3) 2–3 times per month”, “(4) once per week”, “(5) 2–4 times per week”, “(6) 5–6 times per week”, “(7) once a day”, “(8) twice a day”, and “(9) three times per day”. We re-classified the frequency of peanut intake into three groups “non-intake group (previous group 1)”, “intermittent intake group (groups 2–5)”, “frequent intake group (groups 6–9). Therefore, subjects who rarely consumed peanuts, who had an intake of 0.25 to 4 times per week, and who had an intake of 0.7 to 3 times per day were classified into the non-intake, intermittent intake, and frequent intake groups, respectively. Intake of energy and nutrients was calculated using the results of the FFQ. To identify dietary patterns, dietary data were categorized into 13 groups (Appendix A).

### 2.3. Statistical Analysis 

KNHANES uses a complex and multistage clustered probability design to select participants who represent the non-institutionalized Korean population. Survey weights take into account non-responder, over-sampling, post-stratification, and sampling error. Therefore, we applied sample weights to all analyses based on the complex survey design. Normally distributed continuous variables were compared between non-intake, intermittent intake, and frequent intake group using one- way analysis of variance (ANOVA), while the chi-square test was used for categorical variables. Principal components analysis (PCA) was used to derive dietary patterns and to determine factor loadings for each of the 13 food subgroups except peanuts. For proper use of factor analysis, sample homogeneity was tested by examining the distribution of variables in a loading plot, contrasting the value observed against those expected in a normal distribution. This was verified by the Kaiser–Meyer–Olkin (KMO) measurement of adequacy. A KMO value greater than 0.50 was considered acceptable. Varimax rotation was performed to enhance interpretability and to maintain uncorrelated factors. Foods with factor-loading values ≥0.40 were considered to make important contributions to the specific pattern. Dietary patterns were classified according to the highest loading factor: “Prudent”, “Imprudent”, and “Sugar-rich”. Each individual had a separate score for each dietary pattern. The primary outcome was the frequency of peanut intake, as divided into three groups. Because this dependent variable was ordered but not continuous, we used multivariable ordered logistic regression to identify the factors associated with peanut intake. Adjusted odds ratios (aOR) from ordered logistic regression models are presented with 95% confidence intervals. STATA version 16 (StataCorp LLC, Texas, USA) was used for the statistical analysis. A *p*-value less than 0.05 was considered significant.

## 3. Results

Included participants demonstrated a mean (SE) age of 40.9 (1.4) years, and the proportion (SE) of men was 50.1 (1.0) percent. Of the 17,625 subject study population, 10,552 (59.9%), 6726 (38.2%), and 347 (1.9%) subjects were categorized into a non-intake group, an intermittent intake group, and a daily intake group, respectively. The weighted mean (SE) age in non-intake, intermittent intake, and frequent intake groups was 39.1 (0.2) years, 43.5 (0.2) years, and 48.6 (0.8) years, respectively (all *p* < 0.001). 

### 3.1. Sociodemographic Characteristics of Study Participants 

Table 1 shows the sociodemographic characteristics of the study population by the peanut intake group. The weighted proportion (SE) of the non-intake group in men and women was 57.3 (0.7)% and 65.1 (0.6)%, respectively; the weighted proportion (SE) of frequent intake group in men and women was 1.4 (0.1)% and 1.8 (0.1)%, respectively. While the weighted proportion of the non-intake group decreased with older age, higher household income, and higher levels of education, the weighted proportion of the frequent intake group increased with older age, higher household income, and a higher level of education (all *p* < 0.001). For household composition, the weighted proportion of non-intake in a single generation (60.2%) was lower than that in multiple generations (62.6%), but the weighted proportion of frequent intake in a single generation (1.9%) was similar to that in multiple generation households. The weighted proportion of non-intake in obese (65.7%) individuals was lower than that in underweight (71.7%) individuals, but the weighted proportion of frequent intake in obese (1.3%) individuals was higher than that in underweight (1.5%) individuals (*p* < 0.001). There was no significant difference in the areas of residence among the three groups (*p* = 0.532). There was no significant difference in peanut ingestion between pregnant and non-pregnant women between 20 to 40 years old (*p* = 0.665). The proportion of CVD or DM showed an increasing tendency with higher peanut intake (*p* < 0.001). Appendix A shows the proportion of each CVD and DM by the peanut intake group. Characteristics of sociodemographic factors stratified by sex are shown in Appendix A.

As shown in Figure 2A, the weighted percentage of the non-intake group increased over time (59.8% in 2012, 58.1% in 2013, 59.3% in 2014, 63.4% in 2015, and 65.3% in 2016), and those of the daily intake group was relatively constant (1.3% in 2012, 1.7% in 2013, 1.8% in 2014, 1.7% in 2015, and 1.7% in 2016) (*p* for trend <0.001). While the weighted percentage of the non-intake group decreased with older age (74.2% in aged 20–29, 66.4% in aged 30–39, 56.4% in aged 40–49, 50.2% in aged 50–59, and 52.5% in aged 60–69), the percentage of those in the daily intake group increased with older age (0.8% in aged 20–29, 0.8% in aged 30–39, 1.0% in aged 40–49, 3.1% in aged 50–59, and 4.6% in aged 60–69) (*p* for trend <0.001; Figure 2B).

### 3.2. Dietary Pattern of Study Participants 

Factor analysis revealed three main dietary patterns for the study subjects; food groups and factor loadings with absolute values are presented in Table 2. A PCA plot of each three components is presented in Appendix A. Prudent dietary pattern was characterized by high intakes of fruits and vegetables, seafood, and legumes. Imprudent dietary pattern was identified by high intakes of beef, pork, poultry, and refined grains, and low intake of whole grains. Sugar-rich dietary pattern was characterized by high intakes of sweets and beverages. 

The mean (SE) of total energy, macronutrients (carbohydrates, fats, proteins), iron, and fiber intake increased with higher peanut intake (all *p* < 0.001; Table 3). The mean (SE) values of vitamin A, vitamin B1, vitamin B2, vitamin B3, and vitamin C in the peanut frequent intake group were higher than those in the peanut non-intake group (all *p* < 0.001). The weighted proportion of the highest quartile of prudent dietary pattern scores increased according to a higher frequency of peanut intake. (*p* < 0.001; Figure 3). Conversely, the weighted proportion of the highest quartile of imprudent dietary pattern decreased with higher peanut consumption (all *p* < 0.001). The dietary characteristics of the study population stratified by gender are described in Appendix A.

### 3.3. Sociodemographic and Dietary Factors Associated with Frequent Peanut Consumption in Korean

The association between sociodemographic and dietary factors and frequent peanut consumption are presented as aOR and 95% CI (Table 4). After adjusting for age, sex, BMI, history of CVD or DM, level of education, household income, household composition, type of occupation, alcohol consumption, smoking status, and dietary pattern score (Model 3), participants with older age (aOR 1.03; 95% CI 1.03–1.04), higher education (high school graduates: aOR 1.75, 95 CI 1.39–2.19; higher than college: aOR 2.11, 95% CI 1.65–2.70), and skilled work (aOR 1.17; 95% CI 1.01–1.35) were associated with frequent peanut intake. Subjects with CVD or DM did not show a significant association with higher peanut consumption (all *p* < 0.05). As the elderly were more likely to have CVD or DM, we conducted a history of CVD- or DM-stratified ordered logistic regression analysis (Appendix A). Regardless of a history of CVD or DM, age was the independent factor of high peanut consumption. Current smokers were less likely to consume peanuts (aOR 0.73; 95% CI 0.61–0.87). When the association between prudent dietary scores in quartile and peanut intake was analyzed, after controlling for the same confounders, participants with prudent dietary scores in the second (aOR 1.71; 95% CI 1.47–1.99), third (aOR 2.53; 95% CI 2.16–2.97), and fourth quartiles (aOR 3.72; 95% CI 3.16–4.40) showed increased odds of high peanut intake. 

## 4. Discussion

### 4.1. Peanut Consumption in Korea

To the best of our knowledge, this is the first study reporting the sociodemographic and dietary patterns associated with peanut consumption in Korean. We found that 59.8% of Koreans rarely intake peanuts and only 1.9% of the study participants had daily peanut consumption. There was a significant increasing trend in peanut non-intake by year. Overall nutrients including energy, protein, fat, vitamin, and fiber in the diets of the frequent peanut intake group were significantly higher than those in the peanut non-intake group. Interestingly, older age, male, higher education level, and prudent dietary pattern were significantly associated with daily peanut consumption. There was little evidence that peanut consumption was associated with a history of CVD or DM. 

In the present study, high education level and prudent dietary pattern rather than household income or occupation were associated with frequent peanut consumption. Peanuts are known to have cardio-protective, anti-inflammatory, and antioxidant properties that may influence blood lipid level, endothelial function, and inflammatory biomarker [6,38,39]. Therefore, peanuts are a key food component in healthy dietary patterns, such as the Dietary Approaches to Stop Hypertension (DASH) diet or the Mediterranean diet [40]. In addition, a number of studies have demonstrated that peanut consumption was associated with decreased overall [41,42] and CVD-specific mortality [8,10,43]. Liu et al. found that those consuming peanuts seven or more times per week had a 20% lower mortality [36]. Because Koreans consume peanuts as snacks rather than as main foods [44,45], the frequency of peanut consumption may depend on health-related information that can be easier to obtain for better-educated populations. Moreover, the highest prudent dietary pattern scores were increased with high frequency of peanut consumption in our results. This finding is consistent with previous studies that showed a positive correlation between healthy dietary patterns and high levels of education [46,47,48,49]. Finger et al. [46] showed that adults with a low level of education were more likely to consume energy-dense foods and diets low in fruits and vegetables. An Australian study reported that higher education levels and favorable lifestyle characteristics could predict an increase in healthy dietary patterns over four years [48]. In addition, current smokers did not significantly consume peanuts in this study. Health-promoting behavior may be associated with frequent peanut consumption for highly educated household members in Korea.

Frequent peanut consumption was also associated with high energy, macronutrients, fiber, and vitamins intake in our study. These results are consistent with previous studies which aimed to compare diet quality between peanut consumers and non-consumers [50,51,52]. A USA study showed that peanut consumers had higher intakes of protein, total fat, polyunsaturated fat, monounsaturated fat, fiber, vitamin A, and iron [50]. A New Zealand study found that whole nut consumers had higher intakes of dietary fiber, vitamins, and unsaturated fats than those not consuming nuts [51].

One concern associated with frequent peanut consumption is weight gain due to high protein and oil contents. In our study, the frequent peanut intake group had higher energy, carbohydrate, protein, and total fat intake. However, our findings showed that overweight or obese individuals were not associated with frequent peanut consumption when controlling for confounders. Moreover, previous trials found that increased peanut intake was associated with a low risk of obesity [53,54] and decreased weight gain [55,56].

Our results showed that 59.8% of Koreans, the proportion of which continues to increase, were included in the peanut non-intake group. We propose several hypotheses for this finding. First, response bias may have influenced our results because the assessment of peanut consumption was based on self-reports that only asked for “peanut consumption.” This information did not include peanut consumption from peanut butter, peanut flour, peanut cooking oil, and confectionary peanut products. Therefore, peanut consumption may have been underestimated in our study. Second, responders may not have differentiated between tree nuts and peanuts, which may be perceived by the general population to be “nuts”. The peanut is technically categorized as a pea and belongs to the family (*fabaceae*) of beans/legumes [3]. Most previous studies based on surveys investigated peanuts and tree nuts as one item [57,58] because these provide similar nutrition components and are similarly consumed as a processed type of food or as a garnish. Studies for peanut and nut consumption in Koreans are rare. A Korean case–control study for evaluating the association between nut consumption (including peanuts) and colorectal cancer showed that 34.7% of participants were in the nut non-intake group and only 10.5% of participants consumed more than 3 nuts servings per week [59]. However, study participants herein were patients with cancer for whom a healthy dietary pattern was prescribed. Further investigation is needed to evaluate peanut and tree nut consumption status separately in Korean populations.

### 4.2. Perspectives for Peanut Allergy

Recent studies suggested that environmental factors in early life may be important key risk factors for peanut allergy development [60,61]. An Australian study showed that Australian-born Asians had a higher prevalence of nut (including peanut) allergies than Asians who had postnatally migrate from Asia [62]. In addition, high education level was associated with peanut and nut allergy [63,64]. On the other hand, Fox et al. [36] found that household peanut consumption (quantified as the combined peanut consumption of all household members) during the first year of life increased the risk of peanut allergy in atopic children. Brough et al. [65] found that household peanut consumption showed positive correlation with peanut protein presence in both bedroom and living room dust and kitchen surfaces. Trendelenburg et al. [66] showed a positive correlation between household peanut consumption and peanut level in the eating domain. If household members intake peanuts frequently, hands and saliva may be potential sources for direct or indirect contact with an infant’s disrupted skin barrier (due to eczema, filaggrin mutation, or increased transepidermal water loss) [67]. As mentioned earlier, peanuts are not frequently used in traditional introduction of foods in Korean infants. However, especially with infants who live with a household member who consume peanuts frequently, efforts to reduce peanut contact via a skin route or to apply active moisturizer or control skin inflammation may be needed to prevent peanut allergy. Our findings provide information on frequency of peanut consumption in Koreans. Further prospective studies are needed to confirm whether frequent household peanut consumption in household members is correlated with prevalence of peanut allergy in Korean children.

### 4.3. Study Strengths and Limitations

The present study has several limitations. Our results could not determine a causal relationship between sociodemographic characteristics and peanut consumption because of the cross-sectional study design. In addition, the classification of foods into 13 groups, including Korean foods, was an arbitrary decision. However, the dietary patterns identified in our study were similar to those established in previous studies using similar statistical analysis [68,69,70]. Despite these limitations, this is the first study to evaluate the characteristics of peanut consumption in a large Korean population sample.

## 5. Conclusions

We found that 59.8% of Koreans rarely consume peanuts. Frequent peanut intake is associated with high level of education and a prudent dietary pattern in Koreans. This information may be helpful not only in public health research for nutrition, but also in personal management for prevention of peanut allergy in Korea.

## Figures and Tables

**Figure 1 nutrients-12-01207-f001:**
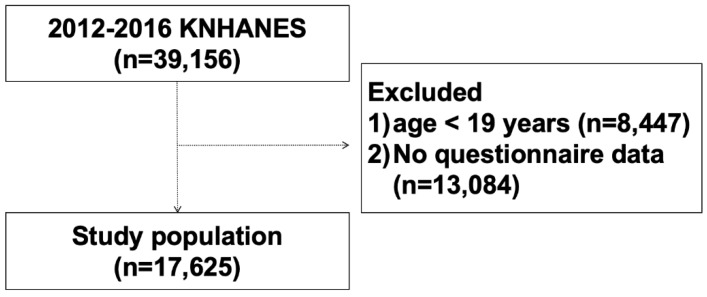
Flow chart showing the study population. KNHANES, Korea National Health and Nutrition Examination Survey.

**Figure 2 nutrients-12-01207-f002:**
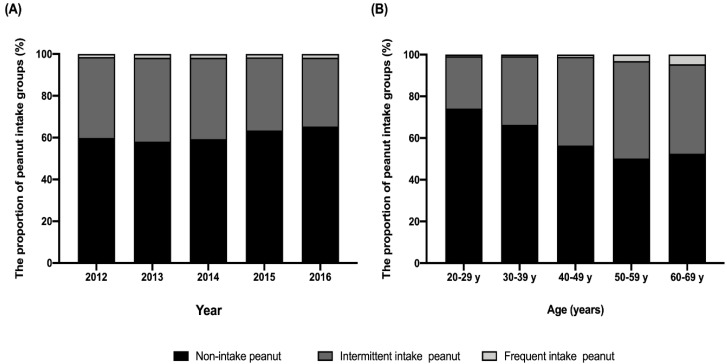
Trend in peanut intake by year and age groups in Koreans. (**A**) The weighted proportion of the non-intake group increased by year, and that of the frequent intake group was relatively constant (*p* for trend <0.001). (**B**) The weighted percentage of the non-intake group showed a declining trend based on older age, while that of the frequent intake group increased with older age (*p* for trend <0.001).

**Figure 3 nutrients-12-01207-f003:**
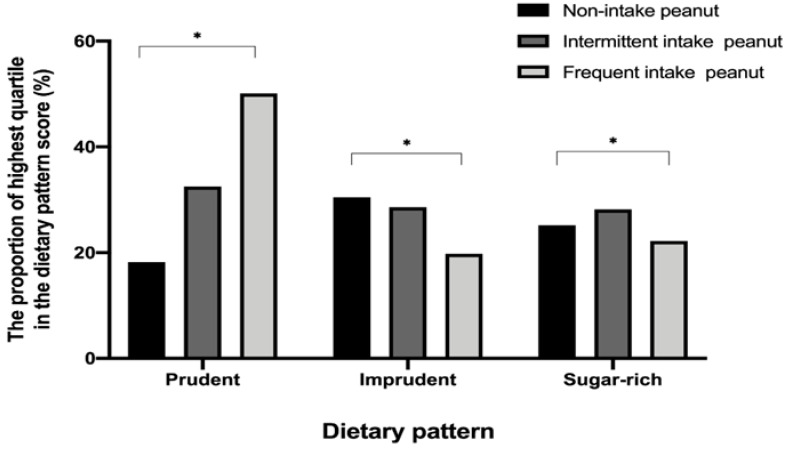
The weighted proportion of the highest quartile of dietary pattern score by peanut intake groups. Study subjects who rarely consumed peanuts, who consumed peanuts 0.25 to 4 times per week, and who consumed peanuts 0.7 to 3 times per day were classified into a non-intake group, an intermittent intake group, and a frequent intake group, respectively. * *p* < 0.05.

**Table 1 nutrients-12-01207-t001:** Sociodemographic characteristics of study participants according to frequency of peanut consumption. (*n* = 17,625).

Variable	Non-Intake (*n* = 10,552)	Intermittent Intake (*n* = 6726)	Frequent Intake (*n* = 347)	*p*-Value *
**Sex**				<0.001
Male	57.3 (0.7)	41.3 (0.7)	1.4 (0.1)	
Female	65.1 (0.6)	33.1 (0.6)	1.8 (0.1)	
**Age group**				<0.001
20–29 years	74.2 (1.0)	24.9 (1.0)	0.8 (0.2)	
30–39 years	66.4 (1.0)	32.8 (0.9)	0.8 (0.2)	
40–49 years	56.4 (1.0)	42.5 (1.0)	1.0 (0.2)	
50–59 years	50.2 (1.0)	46.7 (0.9)	3.1 (0.3)	
60–69 years	52.5 (1.3)	42.9 (1.2)	4.6 (0.5)	
**Resident area**				0.532
Rural	61.2 (0.6)	37.1 (0.5)	1.6 (0.1)	
Urban	60.5 (1.3)	37.9 (1.2)	1.7 (0.3)	
**Household income**				<0.001
Lowest	68.3 (1.6)	30.1 (1.5)	1.6 (0.3)	
Middle–low	64.1 (0.9)	34.5 (0.9)	1.4 (0.2)	
Middle–high	60.9 (0.9)	37.7 (0.8)	1.4 (0.2)	
Highest	57.3 (0.8)	40.7 (0.8)	2.0 (0.2)	
**Household composition**				<0.001
Living alone	60.2 (1.0)	37.9 (1.0)	1.9 (0.2)	
Single generation household	61.2 (0.6)	37.3 (0.6)	1.5 (0.1)	
Multigeneration household	62.6 (1.6)	35.5 (1.6)	1.9 (0.4)	
**Education level**				0.001
<Elementary	62.3 (1.5)	35.0 (1.5)	2.7 (0.4)	
Middle school	60.1 (1.6)	37.8 (1.6)	2.1 (0.4)	
High school	63.0 (0.8)	35.4 (0.8)	1.6 (0.2)	
College or higher	59.4 (0.8)	39.2 (0.8)	1.4 (0.2)	
**Occupation**				<0.001
Unemployed	65.9 (0.8)	39.5 (1.1)	1.6 (0.2)	
Unskilled workers	61.9 (1.6)	40.0 (0.7)	1.2 (0.3)	
Non-manual, skilled workers	58.5 (0.7)	36.9 (0.7)	1.6 (0.2)	
Professionals and managers	58.9 (1.2)	32.4 (0.8)	1.6 (0.2)	
**Smoking status**				<0.001
None	63.1 (0.6)	35.1 (0.6)	1.8 (0.1)	
Former	53.3 (1.1)	45.1 (1.1)	1.5 (0.2)	
Current	63.5 (1.0)	35.2 (1.1)	1.4 (0.2)	
**Alcohol consumption**				0.001
None	63.0 (1.2)	34.8 (1.2)	2.1 (0.3)	
Moderate	60.1 (0.7)	38.4 (0.7)	1.5 (0.2)	
Heavy	60.3 (2.2)	37.8 (2.2)	1.8 (0.5)	
**Body mass index,**				<0.001
Underweight (<18.5 kg/m^2^)	71.7 (1.9)	26.8 (1.8)	1.5 (0.6)	
Normal (18.5–24.9 kg/m^2^)	61.3 (0.6)	37.1 (0.6)	1.7 (0.1)	
Overweight (25.0–29.9 kg/ m^2^)	58.2 (0.9)	40.3 (0.9)	1.6 (0.2)	
Obese (≥30.0 kg/m^2^)	65.7 (2.0)	33.0 (2.0)	1.3 (0.4)	
**History of cardiovascular disease ^†^ or diabetes mellitus**	16.2 (15.2)	20.6 (0.7)	27.5 (3.0)	<0.001

Note: All values are presented as weighted percentages and standard errors. Study subjects who rarely consumed peanuts, who consumed peanuts 0.25 to 4 times per week, and who consumed peanuts 0.7 to 3 times per day were classified into a non-intake group, an intermittent intake group, and a frequent intake group, respectively. * Chi-square test was used to estimate the differences in sociodemographic factors by peanut intake groups. ^†^ Cardiovascular disease was defined as at least one of hypertension, ischemic heart disease, hypercholesterolemia, or stroke. *p* < 0.05 is considered significant.

**Table 2 nutrients-12-01207-t002:** Principal components analysis. Varimax-rotated food group factor loading scores for dietary intake of study participants.

Food Group	Prudent	Imprudent	Sugar-Rich
Beef, pork and poetry	0.184	0.445 *	−0.007
Dairy product	0.228	0.168	−0.133
Fruits and vegetables	0.462 *	0.010	−0.072
Refined grain	−0.025	0.602 *	0.043
Whole grain	0.217	−0.553 *	−0.024
Sweets	−0.003	−0.030	0.680 *
Seafoods	0.417 *	0.005	0.042
Legume	0.423 *	−0.008	0.037
Egg	0.274	0.215	−0.106
Kimchi	0.294	−0.159	0.155
Sugar beverages	0.026	0.033	0.664 *
Alcohol	0.004	0.178	0.182
Seaweed	0.371	−0.005	0.016

* Food group with factor-loading values ≥0.40 were considered to have major contributions to the specific pattern.

**Table 3 nutrients-12-01207-t003:** Dietary characteristics of the study population according to the frequency of peanut consumption.

Variable	Non-Intake (*n* = 10,552)	Intermittent Intake (*n* = 6726)	Frequent intake (*n* = 347)	*p*-Value *
Total energy intake, kcal/d	2041.6 (10.2)	2287.3 (13.6)	2340.0 (60.6)	<0.001
Carbohydrate intake, g/day	317.4 (1.4)	352.6 (1.9)	354.9 (8.4)	<0.001
Protein intake, g/day	66.65 (0.4)	76.9 (0.6)	80.5 (2.3)	<0.001
Total fat intake, g/day	41.7 (0.3)	48.1 (0.4)	55.8 (1.9)	<0.001
Polyunsaturated fatty acid, g/day	10.7 (0.1)	12.8 (0.1)	15.9 (0.5)	<0.001
Monounsaturated fatty acid, g/day	12.95 (0.1)	15.0 (0.2)	18.6 (0.7)	<0.001
Saturated fatty acid, g/day	12.63 (0.1)	13.9 (0.1)	14.81 (0.6)	0.039
Cholesterol, mg/day	262.1 (2.3)	292.8 (3.0)	291.9 (14.7)	0.002
Fiber, g/day	19.1 (0.1)	23.8 (0.2)	28.1 (0.7)	<0.001
Iron intake, g/day	13.3 (0.1)	15.9 (0.1)	17.0 (0.4)	<0.001
Vitamin A RAE, µg/day	594.4 (4.1)	715.3 (5.3)	824.4 (25.0)	<0.001
Vitamin B1, mg/day	1.8 (0.1)	2.1 (0.1)	2.2 (0.1)	<0.001
Vitamin B2, mg/day	1.3 (0.1)	1.5 (0.1)	1.6 (0.1)	0.001
Vitamin B3, mg/day	13.2 (0.1)	15.9 (0.1)	19.3 (0.5)	<0.001
Vitamin C, mg/day	102.7 (1.1)	131.4 (1.3)	176.7 (7.1)	<0.001

* Chi-square test was used to estimate the differences in dietary factors by peanut intake groups.

**Table 4 nutrients-12-01207-t004:** Sociodemographic and dietary factors associated with frequent peanut consumption in Korean.

Variable	Model 1 aOR (95% CI)	Model 2 aOR (95% CI)	Model 3 aOR (95% CI)
**Age**	1.03 (1.02–1.04) *	1.03 (1.02–1.04) *	1.03 (1.03–1.04) *
**Male**	1.42 (1.33–1.52) *	1.42 (1.32–1.53) *	1.67 (1.43–1.95) *
**BMI**			
Underweight (<18.5 kg/m^2^)	Reference	Reference	Reference
Normal (18.5–24.9 kg/m^2^)	1.16 (0.96–1.42)	1.18 (0.95–1.46)	1.11 (0.80–1.54)
Overweight (25.0–29.9 kg/ m^2^)	1.16 (0.95–1.42)	1.21 (0.97–1.51)	1.22 (0.87–1.70)
Obese (≥30.0 kg/m^2^)	0.95 (0.74–1.23)	0.98 (0.74–1.30)	1.07 (0.72–1.61)
**Education**			
≤Elementary school	Reference	Reference	Reference
Middle school	1.26 (1.05–1.50) *	1.28 (1.06–1.55) *	1.27 (0.97–1.66)
High school	1.82 (1.56–2.12) *	1.83 (1.56–2.15) *	1.73 (1.38–2.16) *
≥college	2.14 (1.83–2.50) *	3.11 (1.78–2.50) *	2.10 (1.63–2.65) *
**Household income**			
Lowest	Reference	Reference	Reference
Middle–low	1.30 (1.09–1.55) *	1.33 (1.10–1.61) *	1.16 (0.90–1.48)
Middle–high	1.50 (1.26–1.77) *	1.54 (1.28–1.85) *	1.16 (0.91–1.49)
Highest	1.71 (1.45–2.02) *	1.74 (1.46–2.08) *	1.11 (0.87–1.41)
**Household composition**			
Living alone	Reference	Reference	Reference
Single generation household	1.12 (1.02–1.24) *	1.10 (0.99–1.22)	0.94 (0.82–1.07)
Multigeneration household	1.04 (0.90–1.21)	1.01 (0.86–1.19)	0.93 (0.75–1.17)
**Occupation**			
Unemployed	Reference	Reference	Reference
Unskilled workers	0.89 (0.77–1.04)	0.87 (0.74–1.02)	1.06 (0.85–1.32)
Non-manual, skilled workers	1.12 (1.03–1.23) *	1.11 (1.01–1.23) *	1.17 (1.01–1.35) *
Professionals and managers	1.26 (1.12–1.42) *	1.22 (1.07–1.39) *	0.97 (0.80–1.17)
**History of cardiovascular diseases ^†^ or diabetes mellitus**	1.0 (0.85–1.17)	1.02 (0.02–1.03)	1.01 (0.86–1.19)
**Alcohol consumption**			
None	Reference	References	Reference
Moderate	1.15 (1.02–1.31) *	1.20 (1.04–1.38) *	1.24 (1.07–1.43) *
Heavy	0.93 (0.75–1.17)	1.05 (0.82–1.35)	1.10 (0.84–1.43)
**Smoking status**			
None	References	References	Reference
Former	1.02 (0.90–1.15)	1.0 (0.87–1.15)	0.98 (0.82–1.16)
Current	0.75 (0.66–0.84) *	0.72 (0.63–0.82) *	0.73 (0.61–0.87) *
**Prudent dietary pattern**			
1st quartile	Reference	Reference	Reference
2nd quartile	1.73 (1.54–1.93) *	1.75 (1.55–1.99) *	1.71 (1.47–1.99) *
3rd quartile	2.66 (2.49–2.95) *	2.67 (2.38–3.01) *	2.53 (2.16–2.97) *
4th quartile	3.85 (3.44–4.31) *	3.91 (3.45–4.43) *	3.72 (3.16–4.40) *
**Imprudent dietary pattern**			
1st quartile	Reference	Reference	Reference
2nd quartile	1.19 (1.06–1.33) *	1.16 (1.03–1.30) *	0.99 (0.83–1.18)
3rd quartile	1.07 (0.95–1.20)	1.06 (0.94–1.20)	1.06 (0.88–1.28)
4th quartile	1.20 (1.06–1.36) *	1.19 (1.04–1.36) *	1.02 (0.84–1.24)
**Sugar-rich dietary pattern**			
1st quartile	Reference	Reference	Reference
2nd quartile	1.02 (0.92–1.14)	1.04 (0.93–1.17)	1.03 (0.87–1.23)
3rd quartile	1.14 (1.03–1.26) *	1.12 (1.01–1.26) *	1.12 (0.94–1.33)
4th quartile	0.95 (0.85–1.06)	0.95 (0.84–1.07)	0.88 (0.74–1.06)

aOR, adusted odds ratio; CI, confidence interval. Study subjects who rarely consumed peanuts, who consumed peanuts 0.25 to 4 times per week, and who consumed peanuts 0.7 to 3 times per day were classified into a non-intake group, an intermittent intake group, and a frequent intake group, respectively. Model 1 was adjusted for age (years; continuous) and sex. Model 2 was further adjusted for body mass index (BMI, categorical) and presence of cardiovascular disease or diabetes mellitus. Model 3 was further adjusted for levels of education (categorical), household income (categorical), occupation (categorical), current smoking (categorical), alcohol consumption (categorical), and dietary patterns (categorical). ^†^ Cardiovascular disease was defined as at least one of hypertension, ischemic heart disease, hypercholesterolemia, or stroke. * *p* value <0.05.

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
