# Peer review of "Factors Associated with Frequency of Peanut Consumption in Korea: A National Population-Based Study"

_nutrients, 2020, doi:10.3390/nu12051207_

Round 1
Reviewer 1 Report
General comment: The authors reported peanut consumption in Korea and analyzed its association with several variables. The results are interesting and may provide useful information for guiding dietary and food allergy prevention recommendations. However, the manuscript as written is not acceptable for publication. Many grammatical errors and awkward syntax exist throughout the manuscript. The authors may consider referring the manuscript to a native English speaker to revise the language. One major limitation of the manuscript is the disconnection between introduction and discussion. The introduction exclusively described the allergy aspect of peanut consumption and exposure. However, this topic is very limited in discussion. The authors discussed more about the health benefits of peanut consumption, which is not and should be reflected in the introduction. My other suggestions, comments, and questions are listed in specific comments.
Specific comments:
Line 15: “of Koreans”.
Line 17: “the survey”.
Line 21: Spell out “adjusted odds ratio” and “confidence interval” for the first occurrence.
Line 47: “for Korean infants”.
Lines 50-51: Awkward sentence. Please re-phrase it.
Line 61: Did the survey collect food allergy-related information? If yes, the authors should also analyze those data.
Line 75: “variables”.
Line 89: “square of height”.
Line 93: Diabetes should not be defined as a cardiovascular disease.
Line 98: “intake” is a noun.
Lines 149-150: One must be very careful when reporting this type of results. As the authors mentioned, higher peanut intake is associated with an older population, who are more likely to develop a cardiovascular disease.
Table 1: What do the “2nd degree” and “3rd degree” under “Type of family” mean?
Line 181: Please also show a PCA plot to help visualize the results.
Line 212: Please specify that the values reported in Table 4 are AOR followed by CI.
Line 230: This contradicts the statement in lines 149-150.
Line 234: “high quality protein”.
Line 247: “healthy dietary patterns”.
Line 273: Peanut is an oilseed and is very different from pea. The correct family name is Fabaceae.
Reviewer 2 Report
Interesting article but it does not allow to answer all the questions posed, with results from methodological limitations. I think, you should change the titel of this manuscript.
Round 2
Reviewer 1 Report
General comment: The authors addressed most of my questions during revision. There are still grammatical errors and syntax problems. I do not believe e-World Editing provided a good quality service. I will leave it to the editor’s discretion if the language needs further improvement. I have only a few minor comments and suggestions.
Minor comments:
Line 33: Legumes are defined as plants in the Fabaceae family. Therefore, it is redundant to say that peanut is a legume in the Fabaceae family. I suggest the following revisions for this sentence: “Peanut (Arachis hypogaea) belongs to the family of Fabaceae [1].”
Lines 39-40: “Therefore, peanuts are widely consumed in roasted or boiled forms, and also processed into various forms such as peanut butter …”